# Dietary Habits Are Related to Phase Angle in Male Patients with Non-Small-Cell Lung Cancer

Paraskevi Detopoulou [1,2], Theodora Tsiouda [3], Maria Pilikidou [4], Foteini Palyvou [4], Maria Mantzorou [5], Persefoni Perzirkianidou [3], Krystallia Kyrka [3], Spyridon Methenitis [4,6,7], Foivi S. Kondyli [4], Gavriela Voulgaridou [4], Paul Zarogoulidis [8,9,*], Dimitris Matthaios [10], Rena Oikonomidou [11], Maria Romanidou [12], Dimitrios Giannakidis [13] and Sousana K. Papadopoulou [4]

1 Department of Clinical Nutrition, General Hospital Korgialenio Benakio, 11526 Athens, Greece
2 Department of Nutritional Sciences and Dietetics, University of the Peloponnese, 24100 Kalamata, Greece
3 Pulmonary-Oncology Department, 'Theageneio' Cancer Hospital, 54007 Thessaloniki, Greece
4 Department of Nutritional Sciences and Dietetics, International Hellenic University, 57400 Thessaloniki, Greece
5 Department of Food Science and Nutrition, University of Aegean, 81400 Lemnos, Greece
6 Sports Performance Laboratory, School of Physical Education and Sports Science, National and Kapodistrian University of Athens, 15772 Athens, Greece
7 Theseus, Physical Medicine and Rehabilitation Center, 17671 Athens, Greece
8 Pulmonary Department, General Clinic Euromedica Private Hospital, 54645 Thessaloniki, Greece
9 3rd Surgery Department, AHEPA University Hospital, Aristotle University of Thessaloniki, 54636 Thessaloniki, Greece
10 Oncology Department, General Hospital of Rhodes, 85100 Rhodes, Greece
11 Health Center of Evosmos, 56224 Thessaloniki, Greece
12 Dietitian, Adult Eating Disorders Service, Essex Partnership University NHS Foundation Trust, Wickford SS11 7XX, UK
13 1st Department of Surgery, Attica General Hospital "Sismanogleio—Amalia Fleming", 15126 Athens, Greece
* Correspondence: pzarog@hotmail.com

**Abstract: Introduction:** Lung cancer constitutes the most common cause of cancer death. Phase angle (PhA) has been related to lung cancer prognosis, which implies that the identification of dietary or other factors that could predict or modify PhA may have beneficial effects. Nutritional interventions have been linked with positive changes in PhA in certain types of cancer. **Aim:** The present study aimed to investigate the relationships between dietary habits/nutrition and PhA in NSCLC patients. **Methods:** The sample consisted of 82 male patients with non-small-cell lung cancer (NSCLC; stage IV) from the 'Theageneio' Cancer Hospital (Thessaloniki, Greece). Several parameters were assessed, such as body mass index (BMI), lean mass, PhA, Mediterranean diet score (MedDietScore), dietary patterns, smoking, resting metabolic rate, resting oxygen consumption ($VO_2$), ventilation rate, and physical activity. **Results:** According to our results, a dietary pattern rich in potatoes and animal proteins (meat and poultry) was a significant determinant of PhA (B ± SE, *p*: 0.165 ± 0.08, *p* = 0.05) in multiple linear regression models after adjusting for age, smoking, lean tissue, and MedDietScore. **Conclusion:** In conclusion, dietary patterns may affect PhA, suggesting the crucial role of protein in cancer management and the prevention of sarcopenia.

**Keywords:** lung cancer; diet; phase angle; Mediterranean diet; protein

## 1. Introduction

Lung cancer constitutes the most common cause of cancer death [1]. Patients with non-small-cell lung cancer (NSCLC) represent the majority of lung cancer cases (~85%), while patients with the more aggressive small-cell lung cancer (SCLC) type represent ~10–15% of lung cancer cases [2]. The type of treatment seems to affect the undesirable effects of the treatment, with new therapeutic approaches, such as immunotherapy, being

very promising [3]. One previous study in such patients revealed that a combined treatment of immunotherapy and chemotherapy acts effectively as it seems to prolong the survival rate [4].

Both NSCLC and SCLC lung cancer are associated with malnutrition and sarcopenia, which affect survival and the effectiveness of treatments [5]. Body composition analysis, and especially bioelectrical impedance analysis (BIA), can provide data on classic parameters, such as fat mass, lean body mass, etc., and also on patients' phase angle (PhA). PhA is a direct measurement of the human body's cell integrity, has a prognostic value in lung cancer, as has recently been suggested [6], and may change during cancer therapy [7]. More particularly, low values of PhA and sarcopenia are connected to lung cancer [6] and other types of cancer [8] prognosis through various mechanisms, including inflammation and oxidative stress [6]. Indeed, inflammatory markers, such as C-reactive protein, interleukin-6, and tumor necrosis factor-a (TNF-a) have been negatively related to PhA measurements [9–11], while the antioxidant capacity of the plasma has been positively correlated with PhA [11].

In parallel, an altered redox status has been reported in lung cancer patients [12,13]. This oxidative environment may derive from common behaviors, such as smoking and bad nutritional habits, as well as from disease-related factors, such as chronic inflammation, anorexia, and the depletion of antioxidant enzymes [14]. Moreover, a balanced diet containing antioxidants and anti-inflammatory substances may reduce the advancement of lung cancer and enhance tissue regeneration [15]. There are few data sources connecting diet, sarcopenia, and PhA in cancer patients. Indeed, several dietary interventions have been conducted to combat sarcopenia in patients with lung cancer, with mixed results (protein, oral nutritional supplements with omega-3 fatty acids, multimodal interventions, etc.) [6]. Moreover, dietary factors, such as omega-3 fatty acids, have recently been related to PhA measurements [16,17] and nutritional interventions have been found to increase PhA in colorectal cancer patients [18]. However, there are no data on the relationship between diet and PhA measurements as a surrogate of sarcopenia in patients with lung cancer. Therefore, the present study aimed to investigate the relationships between dietary habits and PhA in NSCLC patients.

## 2. Patients and Methods

### 2.1. Patients

In the present study, 82 NSCLC (stage IV) male patients from the 'Theageneio' Cancer Hospital (Thessaloniki, Greece) were included. Our study was approved by our investigational review board ('Theageneio' Cancer Hospital, Thessaloniki, Greece). Written informed consent was given by all patients. All patients were diagnosed and had first-line treatment. The study was carried out in accordance with the Declaration of Helsinki of 1975 (revised in 1983).

### 2.2. Anthropometric Measurements

Weight and height were measured to the nearest 0.1 kg and 0.1 cm, correspondingly, with a digital scale (SECA 769) and a stadiometer (SECA 220). Measurements were taken in light clothing and without shoes. BMI was then calculated as the ratio of weight (kg) divided by height squared (m$^2$). Waist circumference (cm) was measured after a moderate expiration between the superior iliac crest and the lower rib margin in the midaxillary line. Hip circumference (cm) was measured at the level of the buttocks as the maximal horizontal circumference. Waist to hip ratio was then determined.

### 2.3. Body Composition Measurements

Body composition was assessed with the BIA method by using the Bodystat Quadscan 4000, which is a tetrapolar and multiple-frequency device measuring impedance at 5, 50, 100, and 200 kHz. The measurement was performed according to the instructions of the manufacturer by attaching two sensing electrodes to the wrist and ankle and two current

electrodes to the dorsum of the hand and foot (right side of the patient). Measurements were performed in bare feet. Total body fat percentage, total lean mass, total body, and extracellular and intracellular water were estimated by sex-specific equations from the equipment. Phase angle (at 50 Khz) was calculated as follows [18]:

$$\text{Phase angle} = (\text{resistance}/\text{reactance}) \times (180/\pi).$$

### 2.4. Resting Metabolic Rate (RMR) Measurement and Related Parameters

RMR measurement was done with the portable indirect calorimeter Fitmate GS (Cosmed, Rome, Italy). Subjects were asked to lie in a supine position and rest for 20 min in a silent room. Calibration was performed before each measurement. $VO_2$ and ventilation rate were also measured in mL/min and Lt/min, respectively [19].

### 2.5. Dietary Habits

A short 11-item food frequency questionnaire (FFQ) was administered to assess the usual intake of major food groups. The Mediterranean diet score (MedDietScore) was used to assess Mediterranean diet adherence, with appropriate transformations in food portions if needed [20]. The score ranges from 0 to 55, with a greater score indicating higher adherence to the Mediterranean diet.

### 2.6. Physical Activity Habits

Patients filled in the International Physical Activity Questionnaire (IPAQ) questionnaire (short form) [21].

### 2.7. Smoking Habits

The number of cigarettes and the years of smoking were recorded. Then, pack years were assessed as a measure of tobacco exposure for each subject by multiplying a pack of cigarettes per day by smoking years.

### 2.8. Statistical Analysis

Normality was tested with the Kolmogorov–Smirnoff criterion. Normally distributed continuous variables are presented as mean values $\pm$ standard deviation, while skewed variables served as the median and interquartile ranges. Categorical variables are presented as relative frequencies (%). Spearman correlation coefficients were evaluated to test correlations, in order to account for non-linear associations and/or the associations between skewed variables.

Principal component analysis (PCA) was applied to identify the a posteriori dietary patterns and meal patterns. To decide the number of components to be retained from the factor analysis, the eigenvalues that were derived from the correlation matrix of the standardized variables were examined. Components with an eigenvalue greater than 1 were retained for the data analyses. Moreover, the scree plot was used to confirm the previous decision. Based on the principle that the component scores are interpreted similarly to correlation coefficients (i.e., higher absolute scores indicate that the food group contributes most to the construction of the component), the food patterns (4 patterns in total) were defined in relation to those scores of variables that correlated most closely with the component (absolute loading value > 0.45). The orthogonal rotation, using the varimax option, was employed to derive optimal, non-correlated dietary patterns. The information was rotated to increase the representation of each food as a component.

Linear regression models were applied to identify the variables that predict PhA. Briefly, PhA was considered the dependent variable, and various basic, anthropometric and dietary characteristics were considered the independent variables i.e., age, lean mass, MedDietScore, smoking (pack years), and dietary patterns identified by the PCA analysis. In order to distinguish the effects of dietary patterns from adherence to the Mediterranean diet, these variables were entered into the same model.

All reported *p*-values were two-sided (significance level 5%). SPSS v22 software was used for statistical analysis (IBM Corp. Released 2013, Armonk, NY, USA: IBM Corp.). The level of statistical significance was set at 5%. The statistical package for the social sciences (SPSS 18.0 for Windows, Chicago, IL, USA) was used for all the analyses.

## 3. Results

The basic characteristics of patients participating in this study are presented in Table 1. It is noted that only two subjects reported moderate physical activity and the rest of them reported low physical activity. Moreover, the BMI of patients was $26.9 \pm 5$ kg/m², denoting the presence of underweight and overweight patients. The total lean mass was $57.4 \pm 10.6$ kg and the PhA was $5.1 \pm 0.8°$.

**Table 1.** Basic characteristics of participants.

| | Total (*n* = 82) | |
|---|---|---|
| | Mean or Median | SD or 25th–75th |
| Age (years) | 65.8 | 9.1 |
| Pack-years | 75.5 | 47.5–102.5 |
| BMI (kg/m²) | 26.9 | 5.0 |
| Waist circumference (cm) | 105.0 | 96.0–120.0 |
| Hip circumference (cm) | 104.0 | 98.0–111.2 |
| Waist-to-hip ratio | 1.04 | 0.94–1.10 |
| Total body fat (%) | 27.8 | 7.1 |
| Total lean mass (kg) | 57.4 | 10.6 |
| Total body water (%) | 55.6 | 7.4 |
| Extracellular water (%) | 24.2 | 22.1–26.6 |
| Intracellular water (%) | 30.4 | 29.0–32.7 |
| PhA (º) | 5.1 | 0.8 |
| Resting metabolic rate (Kcal) | 1869 | 414 |
| VO$_2$ (mL/min) | 267.7 | 60.7 |
| Ventilation rate (Lt/min) | 9.98 | 2.08 |

Data are presented as mean and standard deviation for normally distributed variables. Otherwise, data are presented as the median and interquartile ranges (25th–75th). SD: standard deviation; BMI: body mass index; PhA: phase angle. Mean and SD are shown for the following variables: age, BMI, total body fat, total lean mass, total body water, PhA, resting metabolic rate, VO$_2$, and ventilation rate. Median and interquartile range are shown for the following variables: pack years, waist circumference, hip circumference, waist-to-hip ratio, extracellular water, and intracellular water (%).

In Table 2, the basic dietary characteristics of the subjects are presented. The median consumption of legumes and fish was once per week, while the median consumption of meat and poultry was twice per week. The median consumption of olive oil, vegetables, and dairy corresponded to one or more portions per day, while fruit consumption was lower. The median and interquartile range of the MedDietScore of the participants was 31 and 29.0–33.0, respectively. The patients reported low alcohol intake (median intake: one portion/week).

Figure 1 and Table S1 in the Supplementary Materials show the factor loadings for the identification of dietary patterns from PCA analysis. Briefly, four dietary patterns were identified that explained 57.4% of the total variance: Pattern 1 was characterized by the high consumption of whole grains, fruits, and vegetables, pattern 2 by a high intake of potatoes, meat, and poultry, and pattern 3 by high olive oil and low alcohol consumption, while pattern 4 was characterized by high legume and fish consumption.

**Table 2.** Dietary characteristics of patients with lung cancer.

| | Total (*n* = 82) | |
| --- | --- | --- |
| | **Median** | **25th–75th** |
| **Whole wheat grains (portions/week)** | 0.0 | 0–6.0 |
| **Potatoes (portions/week)** | 2.0 | 1.0–4.0 |
| **Fruits (portions/week)** | 5.5 | 5.0–10.0 |
| **Vegetables (portions/week)** | 7.0 | 6.0-11.0 |
| **Legumes (portions/week)** | 1.0 | 0.5–2.0 |
| **Fish (portions/week)** | 1.0 | 0.5–2.0 |
| **Meat (portions/week)** | 2.0 | 1.0–3.0 |
| **Poultry (portions/week)** | 2.0 | 1.0–3.0 |
| **Dairy (portions/week)** | 13.0 | 9.0–15.0 |
| **Olive oil (portions/week)** | 7.0 | 7.0–7.0 |
| **Alcohol (portions/week)** | 1.0 | 0–2.0 |
| **MedDietScore** | 31.0 | 29.0–33.0 |

Data are presented as the median and interquartile range (25th–75th) due to skewed variables.

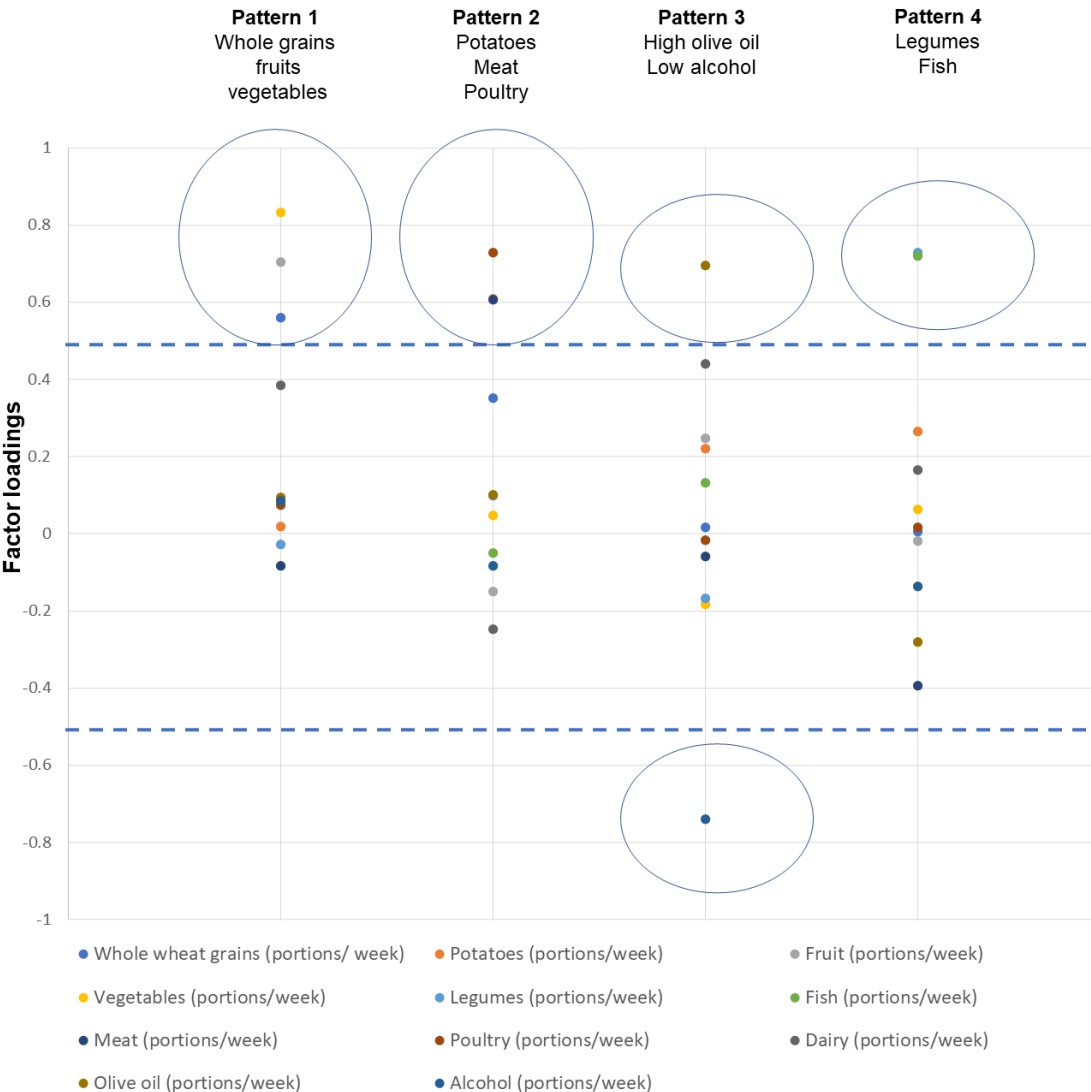

**Figure 1.** Component loadings and identified dietary patterns through principal component analysis.

Higher absolute values of the loadings indicate that the food is correlated with the respective component. Food groups with absolute loadings greater than 0.5 (see dashed

lines) correspond to a specific component. Circles denote the clustering of foods in one component. In total, four components were identified.

Table 3 shows the Spearman correlations between PhA and basic characteristics, anthropometric variables, and the dietary habits of participants. The only significant correlations that were found were between lean mass (kg) (rho = 0.247, *p* = 0.02), food pattern 2 (potato, meat, and poultry; rho = 0.254, *p* = 0.02), and PhA.

**Table 3.** Spearman correlation coefficients between PhA and several parameters.

| | Spearman rho | *p* |
|---|---|---|
| **Waist circumference (cm)** | 0.025 | 0.8 |
| **Hip circumference (cm)** | −0.009 | 0.9 |
| **Waist-to-hip ratio** | 0.071 | 0.5 |
| **Body fat (%)** | −0.147 | 0.1 |
| **Lean tissue (kg)** | **0.247** | **0.02** |
| **Total body water (%)** | 0.021 | 0.8 |
| **Extracellular water (%)** | 0.032 | 0.7 |
| **Intracellular water (%)** | 0.037 | 0.7 |
| **Body cell mass (kg)** | 0.131 | 0.2 |
| **RMR (kcal/day)** | 0.170 | 0.1 |
| **VO$_2$ (mL/min)** | 0.200 | 0.07 |
| **Ventilation rate (L/min)** | 0.07 | 0.5 |
| **MedDietScore** | 0.089 | 0.4 |
| **Food pattern 1: Whole grains, fruits, vegetables** | 0.054 | 0.6 |
| **Food pattern 2: Potato, meat, poultry** | **0.254** | **0.02** |
| **Food pattern 3: High olive oil, low alcohol** | −0.156 | 0.1 |
| **Food pattern 4: Legumes, fish** | 0.129 | 0.2 |

In Table 4, the linear regression models are shown, with PhA as a dependent variable. As can clearly be seen, NSCLC patients' PhA could be better explained/determined by the combination of age, lean mass, and a dietary pattern rich in potatoes, meat, and poultry. The investigated dietary pattern explained PhA independently of age, lean mass, and Mediterranean diet adherence, as expressed by MedDietScore. The total variance explained in the whole model was ~24%.

**Table 4.** Linear regression, with PhA as a dependent variable.

| | Unstandardized Coefficients | | Sig. |
|---|---|---|---|
| | B | Std. Error | |
| **Model 1 (R$^2$ = 23.9%)** | | | |
| *Constant* | 5.572 | 0.752 | 0.000 |
| *Age (years)* | **−0.022** | **0.010** | **0.02** |
| *Smoking (pack years)* | 0.000 | 0.002 | 0.8 |
| *Food pattern 1* | −0.008 | 0.084 | 0.9 |
| *Food pattern 2 (potato, meat, poultry)* | **0.165** | **0.084** | **0.05** |
| *Food pattern 3* | −0.170 | 0.089 | 0.06 |
| *Food pattern 4* | 0.107 | 0.084 | 0.2 |
| *Lean tissue (kg)* | **0.018** | **0.008** | **0.02** |
| **Model 2 (R$^2$ = 24.3%)** | | | |
| *Constant* | 60.387 | 10.382 | <0.0001 |
| *Age (years)* | **−0.023** | **0.010** | **0.02** |
| *Smoking (pack years)* | 0.000 | 0.002 | 0.7 |
| *Food pattern 1* | 0.030 | 0.100 | 0.7 |
| *Food pattern 2 (potato, meat, poultry)* | **0.162** | **0.084** | **0.05** |
| *Food pattern 3* | −0.161 | 0.090 | 0.07 |
| *Food pattern 4* | 0.144 | 0.099 | 0.1 |
| *Lean tissue (kg)* | **0.018** | **0.008** | **0.03** |
| *MedDietScore* | −0.026 | 0.037 | 0.4 |

## 4. Discussion

The main finding of the present study was that a dietary pattern rich in potatoes, meat and poultry seems to determine NSCLC patients' PhA, independently of their age, lean tissue, and MedDietScore.

Low values of PhA have been linked with lung cancer prognosis [6], which implies that the identification of dietary or other factors that could predict or modify PhA may have beneficial prognostic effects. Indeed, several studies report strong correlations between nutrition/nutritional habits and PhA [16,17] Nutritional interventions have been found to increase PhA in colorectal cancer patients [18], while no data exist regarding lung cancer patients.

In the present study, a dietary pattern rich in potatoes and animal proteins (meat and poultry) was positively related to PhA. Interestingly, PhA has recently been reported to be positively associated with meat consumption [22], while it seems to increase after a ketogenic diet independent of weight loss [23]. Of course, in our study, it was mostly a high-protein dietary scheme, rather than a ketogenic pattern that was a significant determinant of PhA. This can be concluded easily since, in the same pattern, high-protein choices (meat, poultry), as well as carbohydrate choices (potatoes), were present. In this context, our results possibly reflect the role of protein sources in cancer management and the prevention of lean mass reduction and sarcopenia [19,24], which, in turn, could influence body composition parameters, such as PhA [6,25].

It is noted that in the present study, no correlation was documented between PhA and other food groups, such as fish, fruits, legumes, and vegetables. This is in contrast with other studies reporting the protective effect of a Mediterranean diet and omega-3 regarding PhA [6,16,17,26]. This discrepancy may be due to the specific features of the present sample. In other words, in disease states, such as cancer, it is possible that the diet-PhA associations are differentiated. As evidenced by our results, food groups recommended for sarcopenia in cancer patients, such as animal protein sources [27], may be more important in determining PhA than a diet rich in antioxidants, which modifies PhA in healthy subjects [26].

The relationship of PhA with age and lean mass has been widely described in the literature. More particularly, PhA decreases with age in both apparently healthy individuals [28–30] and cancer patients [31], reflecting a deterioration in cellular integrity and/or body composition alterations [32], which is in line with our findings. In the present study, PhA was positively related to lean mass, as was expected according to the existing literature [28,30,33]. In the same context, PhA has been positively associated with functional fitness and better physical function in older adults [34], which depends on muscle mass [35].

In the present study, no correlations were observed between PhA and RMR. There are no data on the relationship between PhA and RMR in lung cancer patients. Interestingly, PhA has been found to be a significant predictor of resting energy expenditure in athletes [36], healthy subjects [37,38], and patients with Crohn's disease [39], possibly reflecting the association of PhA with lean mass and muscle quality [40]. This discrepancy may be due to the inherent differences in the studied populations since we included only male subjects with advanced-stage NSCLC. It has also been noted that the proposed prediction equations of RMR, including PhA, constitute more complex functions with additional variables [37,38], so no direct comparison can be made with our results.

The present study included only men. Sex disparities have been documented in lung cancer incidence [41] and sex may also influence treatment outcomes (such as in the case of epidermal growth factor receptor inhibitors and anti-programmed cell-death protein 1 (PD1) inhibitors) [42] or treatment complications [3]. Moreover, there are sex-related body composition [43] and PhA differences [28], and, possibly, sex differences in diet-related behaviors [44]. In this context, the sole inclusion of men can better reflect the net associations of diet with the PhA.

Along with the interpretation of our results, several limitations should be considered. Firstly, etiologic and/or mechanistic conclusions are difficult to draw from the present study since no dietary intervention was performed. Several caveats in the estimation of

dietary intake may have taken place since self-reported data were used. Patients did not provide information about the type and source of meat nor on the way of cooking it. It should also be noted that no biochemical indices were available to assess the inflammatory or nutrient status. Last but not least, we have no data on segmental PhA measurements. However, it has been shown that whole-body PhA is a better predictor of malnutrition in cancer patients when several measurements are compared [31].

In conclusion, the results of the present study suggest that a diet rich in meat, poultry, and potatoes is a determinant of phase angle in male lung cancer patients. A potential change of PhA through diet is of particular importance for the clinical management of lung cancer patients and may have implications for their prognosis. Nevertheless, further well-designed intervention studies are needed to confirm or refute our findings.

**Supplementary Materials:** The following supporting information can be downloaded at: https://www.mdpi.com/article/10.3390/curroncol29110637/s1, Table S1: Component loadings derived using Principal Component Analysis for the identification of dietary patterns

**Author Contributions:** Conceptualization, G.V., D.M., D.G., M.R., T.T., M.M., P.P., K.K., F.S.K., P.Z. and S.K.P.; methodology, T.T., M.M., P.P., K.K., F.S.K., D.M., M.R., R.O., D.G., P.Z. and S.K.P.; formal analysis, S.M. and S.K.P.; investigation, T.T., M.P., F.P., M.M., P.P., K.K., S.M., F.S.K. and D.G.; resources, T.T., M.M., P.P., K.K., D.M., M.R., R.O., D.G., F.S.K. and S.K.P.; data curation, P.D., M.P., F.P. and M.M.; writing—original draft preparation, P.D.; writing—review and editing, S.M., P.Z. and S.K.P.; supervision, P.Z. and S.K.P.; project administration, D.M., M.R., R.O., D.G., T.T., M.P., F.P., M.M., P.P. and K.K. All authors have read and agreed to the published version of the manuscript.

**Funding:** This research received no external funding.

**Institutional Review Board Statement:** The study was approved by the investigational review board of ('Theageneio' Cancer Hospital, Thessaloniki, Greece, Protocol No: 16/07/2018 issue number: 14385) and was performed according to the principles of the Declaration of Helsinki. Written informed consent was acquired from the patients.

**Informed Consent Statement:** Written informed consent was acquired from the patients.

**Data Availability Statement:** The date are available if requested from the corresponding author.

**Conflicts of Interest:** The authors declare no conflict of interest.

## Abbreviations

BIA: bioelectrical impedance analysis; BMI: body mass index; MedDietSCore: Mediterranean diet score; NSCLC: non-small-cell lung cancer; PhA: phase angle; R: resistance; SCLC: small-cell lung cancer; SD: standard deviation; Xc: capacitive reactance.

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
