# Peer review of "Dietary Habits Are Related to Phase Angle in Male Patients with Non-Small-Cell Lung Cancer"

_curroncol, doi:10.3390/curroncol29110637_

Round 1
Reviewer 1 Report
It is my pleasure to help review the manuscrpt. I have some comments and suggestions for authors.
They had a good outcome in the ideas they wanted to convey, subject domain, appropriate statistical methodsused and the way of presenting the results was fine, although they can improved and it could also be presented in graphic form so that the results can be more visible.
For suggestions I might said that in the discussion part comment that they do not have data on segmental PhA measurements but that by comparing several measurements they manage to make a predictor of malnutrition. I would suggest that having the data from the various past measurements, mention the parameters or phase angle cut-offs that were used so the results can have more importance than the one that already haves.
Author Response
Reviewer #1
It is my pleasure to help review the manuscript. I have some comments and suggestions for authors.
They had a good outcome in the ideas they wanted to convey, subject domain, appropriate statistical methodsused and the way of presenting the results was fine, although they can improved and it could also be presented in graphic form so that the results can be more visible.
For suggestions I might said that in the discussion part comment that they do not have data on segmental PhA measurements but that by comparing several measurements they manage to make a predictor of malnutrition. I would suggest that having the data from the various past measurements, mention the parameters or phase angle cut-offs that were used so the results can have more importance than the one that already haves.
We would like to thank the Reviewer for the comments.
We have added a figure (Figure 1) to depict factor loadings and we have kept Table 3 as supplementary information (Supplementary Table 1).
Unfortunately, the authors do not have previous phase angle measurements for these patients.

Reviewer 2 Report
I do not see the corresponding author in the list of authors
methods
- why were only men included in the research? please write why women were not included, what is the significance of this for the study.
Were the patients tested on the BIA without socks? They had bare feet?
is there a scientific source for the phase angle equation? please specify
Table 1 - please indicate which result is the mean and which is the median. Weaving the same SD and 25th-75th
page 6 is almost blank ... please correct it.
Table 3 - what are the results in bold for?
line 187: The word "table 5" without bold
citations should be before the period. It's not good.
Missing: "Author Contributions: "
the references repeats itself. It cannot be like that ... One references is longer, the other is shorter. I guess the correct references is shorter ....
Please correct the references in accordance with the journal's guidelines.
Please standardize the font and its size in accordance with the journal's guidelines.
Author Response
Reviewer #2
- why were only men included in the research? please write why women were not included, what is the significance of this for the study.
The present study included only men. Sex- disparities have been documented in lung cancer incidence and sex may also influence treatment outcomes (such as in the case of epidermal growth factor receptor inhibitors and anti-programmed cell death protein 1 (PD1) inhibitors). Moreover, there are sex-related body composition and PhA differences and possibly sex differences in diet related behaviors. In this context, the sole inclusion of men can better reflect the net associations of diet with the PhA. This point was added at the Discussion with the appropriate references.
Were the patients tested on the BIA without socks? They had bare feet?
Yes, this condition is necessary for the BIA measurement. This point was added at the Methodology
is there a scientific source for the phase angle equation? please specify
Table 1 - please indicate which result is the mean and which is the median. Weaving the same SD and 25th-75th
Table 1 was accordingly updated
page 6 is almost blank ... please correct it.
Done
Table 3 - what are the results in bold for?
Numbers in bold indicate absolute loadings greater than 0.5, which correspond to a specific component.
line 187: The word "table 5" without bold
Done
citations should be before the period. It's not good.
We have updated the manuscript accordingly.
Missing: "Author Contributions: "
We have added this section.
the references repeats itself. It cannot be like that ... e references is longer, the other is shorter. I guess the correct references is shorter ....
We have updated the references list.
Please correct the references in accordance with the journal's guidelines.
Done
Please standardize the font and its size in accordance with the journal's guidelines.
Done
